# Toward Efficient Robust Training against Union of $\ell_p$ Threat Models

**Gaurang Sriramanan, Maharshi Gor, Soheil Feizi**
Department of Computer Science
University of Maryland, College Park
{gaurangs,mgor,sfeizi}@cs.umd.edu

## Abstract

The overwhelming vulnerability of deep neural networks to carefully crafted perturbations known as adversarial attacks has led to the development of various training techniques to produce robust models. While the primary focus of existing approaches has been directed toward addressing the worst-case performance achieved under a single-threat model, it is imperative that safety-critical systems are robust with respect to multiple threat models simultaneously. Existing approaches that address worst-case performance under the union of such threat models (e.g., $\ell_\infty, \ell_2, \ell_1$) either utilize adversarial training methods that require multi-step attacks which are computationally expensive in practice, or rely upon fine-tuning of pre-trained models that are robust with respect to a single-threat model. In this work, we show that by carefully choosing the objective function used for robust training, it is possible to achieve similar, or even improved worst-case performance over a union of threat models while utilizing only single-step attacks during the training, thereby achieving a significant reduction in computational resources necessary for training. Furthermore, prior work showed that adversarial training against the $\ell_1$ threat model is relatively difficult, to the extent that even multi-step adversarially trained models were shown to be prone to gradient-masking and catastrophic over-fitting. However, our proposed method—when applied on the $\ell_1$ threat model specifically—enables us to obtain the first $\ell_1$ robust model trained solely with single-step adversarial attacks. Finally, to demonstrate the merits of our approach, we utilize a modern set of attack evaluations to better estimate the worst-case performance under the considered union of threat models.

## 1 Introduction

Recent years have demonstrated the success of deep learning in solving machine learning tasks spanning across various domains—computer vision, natural language texts, speech, etc. In addition, it has even exceeded the human level performance for certain tasks [He et al., 2016, 2015]. However, despite their successes, these systems exhibit severe vulnerabilities: Deep learning models are very susceptible to imperceptible perturbations in the input at test time [Szegedy et al., 2013]. Such human-imperceptible noise, known as adversarial attacks, can be used to induce networks to confidently predict incorrect labels, and can thus have disastrous implication in safety critical applications such as autonomous navigation and identity verification. To make models robust against such vulnerabilities at test time, a paradigm of *adversarial robust training* of machine learning models has been developed in recent years [Goodfellow et al., 2015, Madry et al., 2018, Zhang et al., 2019].

These adversarial training procedures have primarily been used to train models robust to a single threat model—perturbations constrained within an $\ell_p$-ball of $\varepsilon_p$ radius for some $p$. For instance, the predominant threat model of interest that has been extensively studied in existing literature corresponds to the $\ell_\infty$ threat model (mostly $\varepsilon_\infty = 8/255$). However, human-imperceptible adversarial perturbations can be sourced from multiple threat-models; hence in practice, it is pertinent to ensure that networks are robust against perturbations from a union of threat models simultaneously. More so, it has been observed that robust training procedures for a chosen threat model are not effective against

36th Conference on Neural Information Processing Systems (NeurIPS 2022).

attacks from other threat models [Tramer and Boneh, 2019, Maini et al., 2020], thus necessitating the development of adversarial defenses against multiple perturbation models simultaneously.

Over recent years, training procedures have been proposed to make systems simultaneously robust against perturbations constrained within a union of $\ell_\infty$, $\ell_1$ and $\ell_2$ balls. Systems trained in such manner are then evaluated over the worst-case performance across perturbations from all the threat-models. Tramer and Boneh [2019] proposed simple aggregations of different adversaries for adversarial training against multiple perturbation models utilizing multi-step adversarial attacks for robust training. Maini et al. [2020] further established SOTA for adversarial accuracy against union of ($\ell_\infty$, $\ell_1$, $\ell_2$) perturbations through the adversarial training procedure Multi Steepest Descent (or, MSD) that also uses multi-step ($k = 50$) adversarial attacks to generate adversaries for training.

However, these methods, owing to their requirement of great number of adversarial training steps as compared to a regular setting for multi-step adversarial training procedure (10 steps), are computationally inefficient. This leads to our research question: Is it possible to achieve worst-case performance over a union of threat models that is similar to that of the SOTA methods, while utilizing training procedures that requires only single-step attacks to generate adversaries? We answer the same in affirmation: we first analyse failure modes of existing approaches during $\ell_1$ based adversarial training, and thereby propose to use a dynamic curriculum schedule to effectively mitigate robust overfitting. Furthermore, we extend this approach to develop a training routine that utilizes a single-step adversarial training across a union of threat models to be robust against them simultaneously. In summary, we make the following contributions[1] in this work:

- We demonstrate the first successful single-step robust training procedure, NCAT-$\ell_1$, to achieve $\ell_1$ robustness by using a curriculum schedule with Nuclear Norm based training.

- We extend this approach to propose a training procedure NCAT, that yields SOTA-like robust accuracy under the union of multiple $\ell_p$ threat models, while requiring only a single-step attack budget per minibatch.

- We further demonstrate that the proposed defense can scale-up to high-capacity networks and large-scale datasets such as ImageNet-100. Additionally, NCAT trained models generalize to unseen threat models, achieving near-SOTA robustness even on Perceptual Projected Gradient Descent (PPGD), which comprises one of the strongest attacks known to date.

## 2    Preliminaries

Here, we lay down the notations and conventions used in this work. We denote $x$ to be a $d$-dimensional image from an $N$-class dataset $\mathcal{D}$, while its corresponding ground-truth label as a one-hot vector $y$. $f_\theta$ represents a Deep Neural Network with parameters $\theta$, that maps an input image $x$ to its pre-softmax output $f_\theta(x)$. The cross-entropy loss corresponding to the network prediction on a sample $(x, y)$ is denoted as $\ell_{CE}(f_\theta(x), y)$. For a minibatch $B = \{(x_i, y_i)\}_{i=1}^M$, we denote $X$ as the image matrix whose $i^{th}$ row consists of flattened pixel intensities of the image $x_i$, and $Y$ as the corresponding ground-truth array. Thus, $X$ is a matrix of size $(M \times d)$, and $Y$ is a matrix of size $(M \times N)$. Also, $\ell_{CE}(f_\theta(X), Y)$ now denotes the sum of cross-entropy losses over all data samples in the minibatch $B$. Further, for a matrix $A$, let $\|A\|_*$ denote the Nuclear Norm, the sum of the singular values, of $A$.

**Adversarial Threat Model**: In this work, we primarily consider the robustness of Deep Networks against the union of $\ell_\infty$, $\ell_1$, and $\ell_2$ constrained adversaries. Thus under the $\ell_\infty$ threat model with an $\varepsilon_\infty$-constraint, for a given a clean image $x$, an adversarially perturbed counterpart $\tilde{x}$ can differ by at most $\varepsilon$ at any given pixel location. In contrast, adversaries under the $\ell_1$ constraint may differ from the original image such that the sum of pixel-wise absolute differences are capped by $\varepsilon_1$. Similarly, for adversaries under $\ell_2$ constraint, the sum of squared pixel-wise differences are capped by $\varepsilon_2^2$. Further, a network $f_\theta$ is said to be $\varepsilon_p$-robust under a threat model $\ell_p$ on a clean sample $x$ with label $y$, if $f_\theta(\tilde{x}) = y$, for all perturbations $\tilde{x}$ such that $\|x - \tilde{x}\|_p \leq \varepsilon_p$.

---

[1] Our code and pre-trained models are available here: https://github.com/GaurangSriramanan/NCAT.

## 3 Related Works

In this section we briefly discuss the adversarial attacks and defences that builds up to efficient multi-step adversarial training procedures, work that introduces adversarial training against the union of multiple threat models, and their limitations that we propose to alleviate.

While adversarial training methods have been observed to be the most effective defenses in recent times, early attempts of improving robustness to adversarial attacks included input pre-processing based defenses [Guo et al., 2018, Xie et al., 2018, Song et al., 2018] that were computationally cheap. However, such methods primarily relied upon masking of input gradients in order to counter white-box attacks. Several such defenses of this category were circumvented using smooth approximations of the non-differentiable components, or by utilizing expectation over randomized components [Athalye et al., 2018, Carlini et al., 2019].

### 3.1 Effectiveness of FGSM and its limitations

Perhaps the most successful defense which has stood the test of time is Projected Gradient Descent or PGD adversarial training [Madry et al., 2018]. This involved minimization of cross-entropy loss on the worst-case perturbations generated using multiple iterations of constrained optimization, leading to a significantly higher computational cost when compared to standard training. Multi-step defenses achieve the state-of-the-art robustness today and typically use on the order of 10 steps of optimization for attack generation, with each step requiring an additional forward and backward pass. FGSM or the Fast Gradient Sign Method [Goodfellow et al., 2015] based adversarial training alleviates the computational cost by utilizing single-step adversarial samples for training. However, in practice it is observed that during the course of FGSM training, degenerate solutions are frequently encountered, wherein the local linearity assumption of the loss surface is violated. Indeed, Kurakin et al. [2017] showed that such models exhibited the phenomenon of gradient masking, wherein stronger multi-step attacks were seen to reduce the robust accuracy drastically. Wong et al. [2020] proposed to incorporate early-stopping using R-FGSM based adversarial training [Tramèr et al., 2018], in order to identify the failure-point during robust training with single-step adversaries. However, the method was later shown to not be effective on large capacity networks such as the WideResNet [Zagoruyko and Komodakis, 2016] architecture in subsequent work [Sriramanan et al., 2020].

### 3.2 Nuclear Norm Adversarial Training (NuAT)

Sriramanan et al. [2021] proposes a Nuclear Norm regularizer to improve the adversarial robustness of Deep Networks through the use of single-step adversarial training under $\ell_\infty$ constraints. This Nuclear Norm Adversarial Training (NuAT) enforces function smoothing in the vicinity of clean samples by incorporating joint batch-statistics of adversarial samples, which results in enhanced robustness. Further, this limits the oscillation of function values and prevents the over-smoothing of loss surface uniformly in all directions, leading to a better robustness-accuracy trade-off.

Formally, in a given minibatch $B$, if $X$ is the matrix composed of row-wise vectorized pixel values of each image, $\Delta$ is a matrix of the same dimension as $X$ consisting of independently sampled Bernoulli noise, and $Y$ is the matrix containing the corresponding ground truth one-hot vectors, maximization of the following loss function that utilizes the pre-softmax values $f_\theta(\cdot)$ generates single-step adversaries:

$$\Delta^* = \arg\max_{\Delta} \left[ \ell_{CE}\left(f_\theta(X+\Delta), Y\right) + \lambda \cdot ||f_\theta(X+\Delta) - f_\theta(X)||_* \right] \qquad (1)$$

Subsequently, the following loss function is minimized during Nuclear Norm adversarial training:

$$\min_{\theta} \left[ \ell_{CE}\left(f_\theta(X), Y\right) + \lambda \cdot ||f_\theta(X+\Delta^*) - f_\theta(X)||_* \right] \qquad (2)$$

### 3.3 Union of Threat Models

While above described works trains a target network to be robust against a single threat model of $\ell_p$-ball, there has been recent effort in the direction of making models robust against multiple threat models simultaneously. Tramer and Boneh [2019] study the theoretical and empirical trade-offs of adversarial robustness in various settings when defending against aggregations of multiple

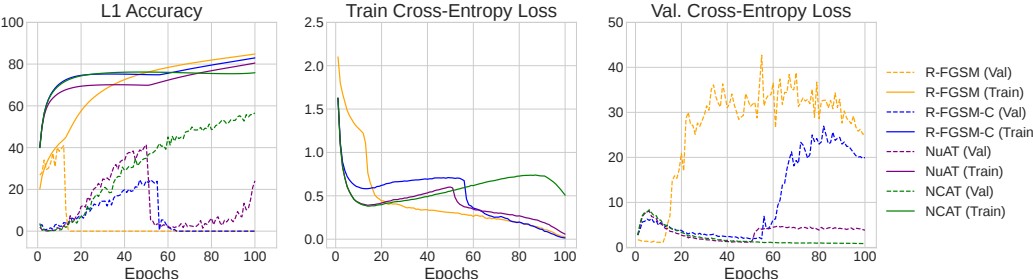

Figure 1: **Catastrophic Overfitting in $\ell_1$ Adversarial Training:** To analyze the stability of single-step training, we plot accuracy (left) and cross-entropy losses (centre, right) over epochs of different single-step adversarially trained models. With R-FGSM based adversarial training [Wong et al., 2020], catastrophic overfitting occurs with extreme gradient masking (orange); adversarial accuracy (loss) is high (low) on the train set, while being close to zero (high) for validation images. More so, even using a curriculum schedule for $\ell_1$ adversaries during training only delays the catastrophic overfitting (blue). In contrast, the proposed training approach NCAT (green) does not display catastrophic overfitting due to gradient masking, and is stable over the entire training regime.

adversaries, proposing to train on the average (AVG) or maximizers of loss (MAX) amongst the different threat models considered for each minibatch of samples. Madaan et al. [2020] train using perturbations generated using a Meta-Noise Generator, and also propose a variant, Stochastic Adversarial Training wherein they utilize multi-step adversaries (10 steps for $\ell_\infty$ and $\ell_2$, 20 steps for $\ell_1$), though the authors note sub-optimal performance from the same. Croce and Hein [2019] propose a provable adversarial defense against all $\ell_p$ norms for $p \geq 1$ using a regularization term for ReLU networks, by enforcing robustness against $\ell_\infty$ and $\ell_1$ adversaries in particular. Lastly and most importantly, Maini et al. [2020] develops a generalization of the standard PGD-based approach to incorporate multiple perturbation models into a single attack by introducing a procedure called Multi Steepest Descent, or MSD, and further utilizing it to train standard architectures that are simultaneously robust against $\ell_\infty$, $\ell_1$ and $\ell_2$ adversaries.

### 3.3.1 Multi Steepest Descent (MSD)

The core idea that MSD [Maini et al., 2020] adopts, which helps establish better worst-case accuracies against the union of adversaries, is to create a single adversarial perturbation by simultaneously maximizing the worst-case loss over all perturbation models at each projected steepest descent step. Unlike previous approaches [Tramer and Boneh, 2019] that generate worst-case adversaries for each threat model, or augment adversaries from multiple threat models, MSD chooses a projected steepest descent direction in each iteration that maximizes the loss over all threat models. This has been established to be superior to the standard adversarial training and the simpler approaches that use comparatively myopic PGD subroutines that only use one perturbation model at a time. However, a clear limitation of MSD here is that it requires 50 adversarial attack steps for each training iteration. Additionally, for each training step, it performs three forward passes (one for each threat model) and a backward pass.

### 3.3.2 Extreme Norms Adversarial Training (EAT)

In order to achieve robustness against a union of $\ell_p$ threat models, Croce and Hein [2021b] propose to fine-tune models that were originally trained to be robust against a single $\ell_p$ norm threat model. The authors demonstrate that fine-tuning of robust models to previously unseen $\ell_p$ threat models is effective, in contrast to adversarial fine-tuning of normally trained networks which yields non-robust models. Furthermore, the authors propose to train solely on $\ell_1$ and $\ell_\infty$ adversaries, such that other $\ell_p$ balls on interest are contained within the union of these two threat models (Extreme Norms Adversarial Training or EAT). However, this can place excessive restrictions during robust training if the perturbation budget of intermediate $\ell_p$ adversaries is large. As with MSD [Maini et al., 2020], EAT can be computationally expensive in practice, since it relies upon multi-step adversarially pre-trained models, and further performs robust fine-tuning of such models using 10-step adversaries in the second phase.

# 4 Proposed Method

As noted by prior works [Madry et al., 2018, Croce and Hein, 2021a, Maini et al., 2020], robust training against the $\ell_1$ threat model is significantly more complicated when compared to standard adversarial training techniques for $\ell_\infty$ or $\ell_2$ threat models. Croce and Hein [2021a] note that even adversarial training using expensive 10-step adversaries generated from SLIDE [Tramer and Boneh, 2019] is prone to catastrophic overfitting [Wong et al., 2020]: Over the course of training, models overfit to the adversaries generated, leading to a false notion of being robust, while achieving close to 0% accuracy against stronger attacks during test evaluation. While such phenomena are frequently seen in single-step training [Goodfellow et al., 2015], the occurrence of such failure modes even with 10-step adversaries exhibits the difficulty in training $\ell_1$ robust networks. Recent work [Wong et al., 2020, Sriramanan et al., 2020, 2021] has demonstrated that such failure modes can be mitigated through appropriate algorithmic choices such as validation-based early stopping using PGD adversaries, or relaxation terms for the overall loss to prevent collapse in training. Croce and Hein [2021a] demonstrate that using the 10-step APGD $\ell_1$ attack, robust models can be trained by automatically tuning the sparsity level induced in the $\ell_1$ perturbations seen during training. Building upon these in this work, we demonstrate the first successful instance of achieving non-trivial $\ell_1$ robustness using single-step adversaries during training. Further, we extend the technique to achieve simultaneous robustness against the union of $\ell_p$ threat models using only single-step training.

## 4.1 Analyzing Robust Training with $\ell_1$ Adversaries

As mentioned before, Croce and Hein [2021a] point out the intricacies involved with $\ell_1$ adversarial training, in that even multi-step training methods can begin to catastrophically overfit. In this work, we seek to identify efficient yet effective single-step training routines that achieve robustness against $\ell_1$ adversaries. In order to build upto that, we first focus on understanding the phenomenon of catastrophic overfitting under this setting and analyze what methods can help alleviate it in the single-step setting.

### 4.1.1 Nuclear Norm Attack and Curriculum Schedule

We begin by plotting the prediction accuracy and cross-entropy loss of different models over training and validation (Figure 1). We find that when trained with R-FGSM based adversaries, models suffer from catastrophic overfitting early on during the training. However, we make a crucial observation that dynamically varying the perturbation budget during training, effectively setting up a *curriculum*, helps immensely in improving overall stability of training. For instance, with the final $\ell_1$ threat model of interest given by the ball of radius 12, we propose to linearly increase this parameter from 0 to 12 to prevent catastrophic overfitting. However, applying this curriculum to RFGSM-AT only leads to a delay in catastrophic failure, indicating the unsuitability of using R-FGSM adversaries for robust training.

Since the goal here is to utilize only single adversarial training step, it becomes imperative that the loss that we optimize over to generate the adversaries is smooth and does not showcase gradient masking. Hence, we build upon Nuclear-Norm Adversarial Training (NuAT) [Sriramanan et al., 2021] to generate single-step adversaries. NuAT proposed to generate $\ell_\infty$-adversaries by maximizing the Nuclear Norm regularized objective. We apply the same for generating adversaries like follows:

$$\widetilde{L} = \ell_{CE}\left(f_\theta(X + \Delta), Y\right) + \lambda \cdot ||f_\theta(X + \Delta) - f_\theta(X)||_* \tag{3}$$

Sriramanan et al. [2021] note that since the Nuclear norm forms a tight convex relaxation for the rank of the predicted matrix of logit values, the corresponding attack generates diverse adversaries in a given minibatch, which then helps mitigate robust overfitting. Crucially, we observe however that this supplemental attack diversity is not sufficient for single-step training on the $\ell_1$ threat model, as even NuAT is observed to be susceptible to catastrophic failure in Fig-1. However, by utilizing the dynamic curriculum schedule, this phenomenon is successfully remedied in the proposed method, NCAT.

### 4.1.2 Steepest Ascent with Single-Step Optimization

Given a minibatch of samples $X$, we generate the Nuclear Norm attack by identifying a perturbation $\Delta$ that maximizes the loss $\widetilde{L}$ as in Eq-3 and conforms to the following two constraints:

$$||\Delta||_1 \le \epsilon_1 , \ X + \Delta \in [0,1]^d$$

wherein the second constraint arises from the normalized range for pixel intensities of an image. Assuming a first-order Taylor series approximation for the loss incurred by the network $f_\theta$, if $\nabla_\Delta \widetilde{L}$ represents the gradient direction, the steepest ascent direction $\Delta^*$ to maximize loss would be parallel to the same in the unconstrained setting. For notational convenience, without loss of generality, consider the optimization problem for an image $x$ of the minibatch $X$. Thus if $g$ represents the corresponding gradient for image $x$ with respect to the loss $\widetilde{L}$, for steepest ascent of loss we have:

$$\max_\delta \left[ \sum_{i=1}^d g_i \delta_i \right] \quad \text{such that} \tag{4}$$

$$\text{(a)} \ 0 \le x_i + \delta_i \le 1 \ \forall \ i, \ \text{and} \ \text{(b)} \ ||\delta||_1 \le \epsilon_1$$

In the absence of constraint (b), the optimal perturbation $\delta$ is given by $\delta = M$ where $M_i$ denotes the deviation budget required at pixel $i$ to saturate the same to the pixel constraint $[0, 1]$, parallel to the gradient $g_i$, and can defined formally as:

$$M_i = \begin{cases} 1 - x_i & \text{if } g_i \ge 0 \\ -x_i & \text{if } g_i < 0 \end{cases}$$

When the overall available budget is limited by $\varepsilon_1$ as in constraint (b), such that $||M||_1 < \varepsilon_1$, the constraint is inactive, and the solution is unaltered. On the other hand, if constraint (b) is active, the solution is necessarily different, wherein a reduction in the perturbation allocated for some pixel locations is made mandatory. Thus, the inner-product in Eq-4 is maximized by assigning the perturbations $M_i$ in priority-order for different pixel locations, based on decreasing magnitude of the absolute gradient values. Thus, if $a_i = |g_i|$ represents the gradient magnitudes at different pixel locations, let $\sigma$ denote the sorted permutation of indices such that $a_{\sigma(1)} \ge a_{\sigma(2)} \ge \cdots \ge a_{\sigma(d)}$. Further, let the cumulative budget utilized be defined as $S_i = \Sigma_{j=1}^i |M_{\sigma(j)}|$ for $i \in \{1, \ldots, d\}$, with $S_0 = 0$. Since each term $|M_{\sigma(j)}|$ in the summand is positive, $S_i$ increases monotonically. Thus with $I_i = \max\{0, \varepsilon_1 - S_{i-1}\}$ denoting support variables for indices which receive a lower perturbation allocation due to constraint (b), the optimal single-step perturbation $\delta^*$ corresponding to image $x$ is then defined as:

$$\delta_{\sigma(i)}^* = \begin{cases} M_{\sigma(i)} & \text{if } S_i \le \varepsilon_1 \\ M_{\sigma(i)} \cdot I_i & \text{if } S_i > \varepsilon_1 \end{cases} \tag{5}$$

Thus, in an $\ell_1$ constrained attack the gradients have to be sorted by their magnitude, which requires $O(d \cdot \log d)$ complexity where $d$ represents the image dimensionality. However, in practice this overhead is observed to be exceedingly minimal relative to data loading times etc. Indeed, using this routine with Nuclear norm based training with the curriculum schedule (NCAT-$\ell_1$), we demonstrate for the first time that single-step training can be effectively used to produce $\ell_1$ robust models.

## 4.2 Scheduling a Curriculum for Robust Training

In order to improve the efficacy of single-step adversarial training, we propose to utilize a dynamically varying perturbation budget during training, to ensure that the model receives supervision on increasingly difficult adversaries as determined by a Curriculum schedule. As remarked previously from Figure-1, robust training using single-step adversaries gains a significant degree of stability with a Curriculum schedule. Different curricula are seen to be effective in practice, including those that incorporate the robust accuracy on training samples observed at a given epoch to dynamically vary the perturbation budget at different rates. However, these complex curricula require additional fine-tuning of hyperparameters, and are sensitive to the $\ell_p$ threat model considered. We thus utilize a simple, linear curriculum wherein the perturbation size is increased from zero to $\varepsilon$, and kept constant for the last few epochs, to facilitate adequate fine-tuned training on the final threat model of interest.

The curriculum schedule proposed here is particularly effective for single-step training given that it does not require sampling of adversarial statistics at different radii. In contrast, prior works largely require an inherent multi-step adversarial generation approach to sample different stopping points to implement a curriculum in practice. For instance, Zhang et al. [2020] proposed Friendly Adversarial Training (FAT), wherein early stopping is performed during the generation of the multi-step PGD attack — the perturbation updates are stopped as soon as the PGD attack successfully induces misclassification on the training image.

**Algorithm 1** Nuclear Curriculum Adversarial Training for $\ell_p$ Norm Robustness

---

1: **Input:** Network $f_\theta$ with parameters $\theta$, Weight Averaged Network $f_\omega$ with parameters $\omega$, Training Data $\mathcal{D}$ with input images of dimension $d$, Minibatch Size M, Attack Size $\varepsilon_p$ for each $\ell_p$ threat model, Epochs $E$, Learning Rate $\eta$, Decision Function D, Curriculum Schedule $\mathcal{C}$

2: **for** $epoch = 1$ **to** $E$ **do**

3:     $\varepsilon_p = \mathcal{C}(p)$

4:     **for** minibatch $\{(x_i, y_i)\}_{i=1}^M \subset \mathcal{D}$ **do**

5:         $X = \begin{bmatrix} \dots & x_1 & \dots \\ \dots & \vdots & \dots \\ \dots & x_M & \dots \end{bmatrix}, \quad \Delta = \begin{bmatrix} \dots & \delta_1 & \dots \\ \dots & \vdots & \dots \\ \dots & \delta_M & \dots \end{bmatrix}, \quad \delta_i \sim Bern^d(-\alpha, \alpha), \quad Y = \begin{bmatrix} y_1 \\ \vdots \\ y_M \end{bmatrix}$

6:         $\widetilde{L} = \ell_{CE}\left(f_\theta(X + \Delta), Y\right) + \lambda \cdot ||f_\theta(X + \Delta) - f_\theta(X)||_*$

7:         **for** $p$ in $\mathrm{D}(\theta)$ **do**

8:             $\Delta = \Delta + \varepsilon_p \cdot \mathrm{Proj}\left(\nabla_\Delta \widetilde{L}, B_p(\varepsilon_p)\right)$

9:             $\widetilde{X} = Clamp\left(X + \Delta, 0, 1\right)$

10:         **end for**

11:         $L = \ell_{CE}(f_\theta(X), Y) + \lambda \cdot ||f_\theta(\widetilde{X}) - f_\theta(X)||_*$

12:         $\theta = \theta - \dfrac{1}{M} \cdot \eta \cdot \nabla_\theta L$

13:         $\omega = \tau \cdot \omega + (1 - \tau) \cdot \theta$

14:     **end for**

15: **end for**

---

## 4.3 Single-Step Training for $\ell_\infty$ and $\ell_2$ Robustness

Since the $\ell_\infty$ threat model is the most well-studied setting in existing literature, we rely upon prior works to obtain excellent baselines. To achieve robustness against $\ell_\infty$ constrained adversaries using single-step training, we utilize the current state-of-the-art method, Nuclear Norm Adversarial Training (NuAT) [Sriramanan et al., 2021]. We further seek to incorporate other threat models during training, in order to obtain models with non-trivial robustness against the union of the $\ell_1, \ell_2$ and $\ell_\infty$ threat models simultaneously. In order to efficiently train against $\ell_2$ adversaries, we first propose to modify the NuAT training algorithm to utilize this constraint set, using $\ell_2$ norm based projections. However, similar to Croce and Hein [2021b], we make the remarkable observation that models that are trained solely on $\ell_\infty$ adversaries achieve a great degree of robustness versus $\ell_2$ adversaries on the test set. We observe similar transfer of robustness from $\ell_1$ trained models toward the $\ell_2$ threat model as well. Thus, the primary difficulty in achieving robustness to the union of threat models appears to be that of training networks robust to the $\ell_1$ and $\ell_\infty$ threat models in particular.

## 4.4 Sampling Procedures to Improve Efficiency

To achieve robustness against the union of the three $\ell_p$ threat models considered, it is plausible that training with three distinct single-step attacks (constrained to $\ell_1$, $\ell_2$ and $\ell_\infty$) using the proposed approach in each minibatch will be effective. However, in this work, we primarily focus on reducing the training complexity further, in order to effectively utilise only a single-step attack for each minibatch. Awasthi et al. [2021] proposed to utilize the multiplicative weights algorithm, wherein the loss under different adversaries on a hold-out validation set guides the sampling procedure, using a set of exponential running weights $w_i$ for each threat model. However, we find that this is contingent on the efficacy of adversaries utilized on the validation set, which can be restrictive in practice due to computational constraints. In practice, we observe that it is indeed sensitive to the degree of convergence achieved by different adversaries, and requires additional tuning for the update coefficient hyperparameters.

Building upon this, we find in practice that alternating between $\ell_\infty$ and $\ell_1$ attacks across different minibatches with a fixed frequency is remarkably effective. Thus, the proposed defense, NCAT, uses nuclear norm based single-step training following a curriculum schedule, such that different threat models are selected for attack generation in different minibatches based on a pre-fixed frequency. We present a concise, summarised overview of the proposed training approaches in Algorithm-

Table 1: **Consolidated Results on CIFAR-10:** Prediction accuracy (%) of ResNet-18 models trained using different methods under various threat models. Robust evaluations are presented under the constraint sets given by the $\ell_1$ ball of radius 12, $\ell_2$ ball of radius 0.5 and $\ell_\infty$ ball of radius 8/255 comprising the individual threat models of interest, along with worst-case and average-case performance under the union of these threat models.

| Method | Number of AT Steps | Clean Acc | Worst-Case Acc | Average Acc | $\ell_1$ Acc | $\ell_2$ Acc | $\ell_\infty$ Acc |
|---|---|---|---|---|---|---|---|
| $\ell_1$ Training Alone | | | | | | | |
| APGD-$\ell_1$ | 10 | 85.9 | 22.1 | 48.8 | 59.5 | 64.9 | 22.1 |
| NCAT-$\ell_1$ | 1 | 81.1 | 37.9 | 53.6 | 55.9 | 67.0 | 38.0 |
| Training under Union of Threat Models | | | | | | | |
| SAT | 13.33[†] | 83.9 | 40.4 | 54.2 | 54.0 | 68.0 | 40.7 |
| AVG | 30 | 84.6 | 40.1 | 53.8 | 52.1 | 68.4 | 40.8 |
| MAX | 30 | 80.4 | 44.0 | 53.4 | 48.6 | 66.0 | 45.7 |
| MSD | 50 | 81.1 | 43.9 | 53.4 | 49.5 | 65.9 | 44.9 |
| EAT | 10[††] | 82.2 | 42.4 | 54.6 | 53.6 | 67.5 | 42.7 |
| NCAT | 1 | 80.3 | 42.6 | 53.3 | 46.9 | 67.0 | 46.0 |
| NCAT[+] | 1 | 77.5 | 43.7 | 53.4 | 48.4 | 65.7 | 46.1 |

1. Here, the Decision Function $D$ (L7, Alg-1) alternately outputs $p = 1$ or $p = \infty$ based on a predetermined frequency, since such models are observed to simultaneously achieve $\ell_2$ robustness without explicit training. As observed in prior works [Chen et al., Sriramanan et al., 2021], maintaining a exponential running average of network weights (SWA [Izmailov et al., 2018]) helps improve robust performance overall as well, particularly so in this setup since different (random) minibatches are trained with adversarial perturbations arising from different threat models. Furthermore, this effectively reducing undesired bias to a particular threat model due to auto-correlations that arise in training. We thus use these exponentially averaged models for final evaluation. A modified version of the proposed approach, namely NCAT-AVG, uses a Decision Function $D$ that outputs the collection of $p = \{1, 2, \infty\}$, and effectively uses a budget of three single-step attacks, one for each threat model. We present results obtained using NCAT-AVG, sampling with multiplicative weights based updates and other ablations in the Supplementary Material.

## 5 Experiments and Analysis

In this work, we primarily consider the CIFAR-10 [Krizhevsky et al., 2009] and ImageNet-100 [Russakovsky et al., 2014] datasets, since they have come to form the benchmark for comparative analysis of adversarially robust models. Following prior works [Maini et al., 2020], we consider constraint sets given by the $\ell_\infty$ ball of radius 8/255, $\ell_2$ ball of radius 0.5 and $\ell_1$ ball of radius 12 as the threat models of interest, and as explained previously, we attempt to train models that achieve non-trivial worst-case accuracy against the union of such $\ell_p$ threat models. For the ImageNet-100 dataset, the corresponding radii for $\ell_1$, $\ell_2$ and $\ell_\infty$ threat models are 255, 1200/255 and 4/255 respectively, following the constraints considered by Laidlaw et al. [2021].

We present results in the white-box setting, wherein the adversary is cognizant of the model weights, architecture and training scheme employed. To accurately estimate worst-case performance, we focus our evaluation pipeline to incorporate state-of-the-art attacks such as AutoAttack [Croce and Hein, 2020] for each $\ell_p$ threat model. Furthermore, AutoAttack includes strong $\ell_1$ attack evaluation baselines using techniques proposed by Croce and Hein [2021a], wherein the authors note that significant improvement in attack efficacy as compared to prior works such as SLIDE Attack [Tramer and Boneh, 2019], B&B Attack [Brendel et al., 2017] and Pointwise Attack [Schott et al., 2018]. Due to paucity of space, we include black-box evaluations, generalization to unseen domains, gradient masking checks and adaptive attacks in the Supplementary Material. We however make the important note that the suite of white-box attacks considered using AutoAttack are *strictly stronger* than the former set of attack evaluations, indicating the absence of gradient masking.

Table 2: **Consolidated Results on CIFAR-10:** Prediction accuracy (%) of WideResNet-28-10 models trained using different methods under various threat models. Robust evaluations are presented under the constraint sets given by the $\ell_1$ ball of radius 12, $\ell_2$ ball of radius 0.5 and $\ell_\infty$ ball of radius 8/255 comprising the individual threat models of interest, along with worst-case and average-case performance under the union of these threat models.

| Method | Number of AT Steps | Clean Acc | Worst-Case Acc | Average Acc | $\ell_1$ Acc | $\ell_2$ Acc | $\ell_\infty$ Acc |
|---|---|---|---|---|---|---|---|
| | | | $\ell_1$ Training Alone | | | | |
| APGD-$\ell_1$ | 10 | 83.7 | 30.7 | 52.5 | 61.6 | 65.1 | 30.7 |
| NCAT-$\ell_1$ | 1 | 80.7 | 39.2 | 54.6 | 56.1 | 68.6 | 39.3 |
| | | | Training under Union of Threat Models | | | | |
| SAT | 13.33[†] | 80.5 | 45.7 | 56.2 | 55.9 | 66.7 | 45.9 |
| AVG | 30 | 82.5 | 45.1 | 56.1 | 55.0 | 68.0 | 45.4 |
| MAX | 30 | 79.9 | 47.4 | 54.6 | 50.2 | 65.3 | 48.4 |
| MSD | 50 | 80.6 | 46.9 | 55.1 | 51.7 | 65.6 | 48.0 |
| EAT | 10[††] | 79.9 | 46.4 | 56.3 | 56.0 | 66.2 | 46.6 |
| NCAT | 1 | 81.5 | 44.6 | 54.8 | 49.9 | 68.3 | 46.3 |

We first present results obtained using the ResNet-18 [He et al., 2016] architecture on CIFAR-10 in Table-1. In the first partition of the table, we present models trained solely on the $\ell_1$ threat model. The current state-of-the-art is achieved by APGD-$\ell_1$ [Croce and Hein, 2021a], which performs a 10-step APGD attack during training in order to mitigate gradient masking and catastrophic overfitting for $\ell_1$ constrained adversaries. On the other hand, our method, NCAT-$\ell_1$ which uses just a single-step attack for adversarial training achieves $\ell_1$ robustness much more efficiently. We note that while the multi-step approach has higher $\ell_1$ robustness (+3.6%), the single-step NCAT trained model has significantly better worst-case (+15.8%) and average-case (+4.8%) accuracy under the union of all three threat models. Indeed, we note once again that NCAT-$\ell_1$ represents the first-ever successful single-step adversarial training on the $\ell_1$ threat model, which also generalizes well to the unseen $\ell_\infty$ and $\ell_2$ threat models simultaneously.

In the second partition of Table-1, we present models that are explicitly trained to be robust under the union of the $\ell_\infty, \ell_2$ and $\ell_1$ threat models. Namely, we present comparative analysis with respect to existing multi-step adversarially trained defenses such as AVG and MAX [Tramer and Boneh, 2019], MSD [Maini et al., 2020], EAT [Croce and Hein, 2021b] and SAT [Madaan et al., 2020]. For these methods, we primarily utilize robust evaluations as presented by Croce and Hein [2021b] to enable fair comparisons, which comprise of re-implemented models that obtain higher accuracies as compared to values reported in the original papers. We first note that SAT[†] requires 13.33 adversarial attack steps during training, since it utilizes 10-step attacks for $\ell_\infty$ and $\ell_2$ adversaries, and 20 attack steps for $\ell_1$ adversaries to mitigate gradient masking, indicating the considerable difficulty involved in achieving $\ell_1$ robustness. In contrast, EAT[††] relies upon 10-step fine-tuning of a network that is already robust against a single threat-model. The current state-of-the-art approaches comprise of MSD and MAX that achieve 44% worst-case accuracy, while utilizing a budget of 50 and 30 attack steps respectively during training. We observe that the proposed approach, NCAT achieves comparable worst-case and average-case performance over the threat models considered, while requiring a significantly smaller computational footprint during training. The relative trade-off between clean accuracy and robustness can be adjusted depending on individual use-case requirements by tuning the coefficient ($\lambda$) of the Nuclear norm regularization term in the overall loss objective. The proposed method facilitates this trade-off, and is presented in the last row as NCAT[+] wherein this coefficient is increased so as to achieve near-SOTA robust performance, at the cost of lower clean accuracy.

In Table-2, we present results on models trained with the WideResNet-28-10 [Zagoruyko and Komodakis, 2016] architecture to demonstrate the scalability of the proposed defense to high-capacity networks. We thus establish the efficacy of the curriculum schedule combined with nuclear norm based training in mitigating catastrophic overfitting, enabling efficient training of these large networks. Similar to previous remarks, we again note that while the multi-step APGD-$\ell_1$ model has higher $\ell_1$

Table 3: **Results on ImageNet-100:** Prediction accuracy (%) of models trained using different methods under various threat models. Robust evaluations are presented under the constraint sets given by the $\ell_1$ ball of radius 255, $\ell_2$ ball of radius 1200/255 and $\ell_\infty$ ball of radius 4/255 comprising the individual threat models of interest, along with worst-case and average-case performance under the union of these threat models.

| Method | Number of AT Steps | Arch | Clean Acc | Worst-Case Acc | Average Acc | $\ell_1$ Acc | $\ell_2$ Acc | $\ell_\infty$ Acc | PPGD Acc |
|---|---|---|---|---|---|---|---|---|---|
| $\ell_\infty$-AT | 10 | RN50 | 81.7 | 0.8 | 20.7 | 0.8 | 3.7 | 55.7 | 1.5 |
| PAT | 10 | RN50 | 72.6 | 37.8 | 41.2 | 41.2 | 37.7 | 45.0 | 29.2 |
| NCAT-$\ell_1$ | 1 | RN18 | 64.9 | 41.1 | 43.9 | 48.3 | 41.4 | 42.1 | 26.6 |
| NCAT | 1 | RN18 | 63.9 | 41.5 | 44.8 | 46.8 | 41.9 | 45.7 | 29.1 |

robustness (+5.5%), the single-step NCAT-$\ell_1$ trained model has improved worst-case (+8.5%) and average-case (+2.1%) accuracy under the union of all three threat models.

In Table-3, we present evaluations on the ImageNet-100 dataset, wherein we utilize ResNet-18 networks to reduce computational demands. We observe that NCAT-$\ell_1$ attains remarkable robust accuracy on unseen $\ell_\infty$ and $\ell_2$ adversaries, even achieving 26.6% accuracy against the Perceptual Projected Gradient Descent (PPGD) attack [Laidlaw et al., 2021], which forms one of the strongest attacks known to date. Further, we observe that the NCAT trained achieves state-of-the-art worst-case $\ell_p$ accuracy, while attaining robustness similiar to that of Perceptual Adversarial Training [Laidlaw et al., 2021] for the PPGD attack, with the latter being a model that was explicitly trained on such adversaries.

## 6 Conclusions

In this work, we develop an efficient adversarial training procedure, NCAT, to train networks that are robust against a union of $\ell_p$ threat models, namely $\ell_\infty$, $\ell_1$ and $\ell_2$. To do so, we first focus on developing an efficient, yet effective robust training procedure for the $\ell_1$ threat model, by incorporating a curriculum schedule to mitigate catastrophic overfitting. Indeed, in this work we present the first $\ell_1$ constrained robust model trained solely using single-step adversaries, achieving robustness similar to that of multi-step SOTA approaches. Furthermore, we extend the proposed method to achieve worst-case robustness under multiple $\ell_p$ norm constraints simultaneously. Compared to the current SOTA that uses 30 adversarial attack steps for its training procedure to achieve 44% robust accuracy on CIFAR-10, our method yields 43.7% robustness while solely utilizing single-step adversaries during the training routine. This thereby greatly reduces the computational requirements needed to achieve SOTA-equivalent robust performance.

## 7 Limitations and Societal Impact

The proposed approach, NCAT, facilitates efficient yet effective training of robust models, that achieve non-trivial worst-case accuracy under the union of several $\ell_p$ norm threat models. Furthermore, NCAT trained models are seen to robust against unseen attacks and adversaries that might potentially be encountered in the real world. Thus, given that NCAT helps train reliable and trustworthy models using a small computational footprint, this work indeed has potential in creating a positive impact on society. We do not see any immediately foreseeable negative consequences associated with our work. Further, in order to surmount existing limitations, we hope that future works could extend NCAT to include provable certificates for different $\ell_p$ threat models and their union, and possibly provide stricter guarantees for the absence of gradient masking and catastrophic failure with the use of theoretically-motivated curriculum schedules.

## 8 Acknowledgements

This project was supported in part by NSF CAREER AWARD 1942230, a grant from NIST 60NANB20D134, HR001119S0026 (GARD), HR00112090132, ONR YIP award N00014-22-1-2271, Army Grant No. W911NF2120076 and the NSF award CCF2212458. We would also like to thank Tianyi Zhou and David Jacobs for fruitful discussions on initial versions of this work.

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
