# Supplementary Material: Toward Efficient Robust Training against Union of $\ell_p$ Threat Models

**Gaurang Sriramanan, Maharshi Gor, Soheil Feizi**
Department of Computer Science
University of Maryland, College Park
{gaurangs,mgor,sfeizi}@cs.umd.edu

## 1 Black-Box and Zeroth-Order Attacks

Table 1: **Black-Box and Unseen attacks on CIFAR-10:** Prediction accuracy (%) of ResNet-18 models trained using APGD-$\ell_1$, NCAT-$\ell_1$ and NCAT. Square Attack evaluations are presented with adversaries lying within an $\ell_1$ ball of radius 12 and $\ell_\infty$ ball of radius $8/255$.

| Method | Number of AT Steps | Clean Acc | Square $\ell_1$ | Square $\ell_\infty$ | Common Corr. | Elastic | Gabor |
|---|---|---|---|---|---|---|---|
| APGD-$\ell_1$ | 10 | 87.1 | 71.8 | 40.8 | 72.0 | 48.7 | 12.4 |
| NCAT-$\ell_1$ | 1 | 81.7 | 65.2 | 48.5 | 67.0 | 54.1 | 12.9 |
| NCAT | 1 | 80.3 | 60.1 | 53.8 | 65.0 | 71.4 | 14.9 |

While white-box attacks that utilize first-order methods generally form the strongest suite of adversarial perturbations, it is plausible that models are not inherently robust, but rather rely upon obfuscated or shattered gradients [Athalye et al., 2018] to falsely display high robust accuracies against such attacks. In this section, we thus present robust evaluations using attack methods that do not rely upon gradient information to craft adversaries.

For Black-box evaluation, we primarily rely on the Square attack [Andriushchenko et al., 2020], since it has been shown to be the strongest gradient-free attack presently. As shown in Table-1, the NCAT and NCAT-$\ell_1$ models achieves significantly higher robust accuracy on the Square Attack as compared to the evaluation presented in Table-1 of the Main paper, indicating that zeroth-order adversaries are weaker than gradient-based attacks. As expected, we also note that the NCAT model trained explicitly on the union of threat models obtains higher Square $\ell_\infty$ accuracy as compared to the NCAT-$\ell_1$ model. On the other hand, for Square $\ell_1$ adversaries, the NCAT-$\ell_1$ model outperforms NCAT by 5%, since training on specific adversaries on a narrow threat model is more efficacious against similar adversaries during test-time. Comparing with the APGD-$\ell_1$ which takes 10 adversarial steps, our approach transfers significantly better over attacks from other threat models:NCAT-$\ell_1$ performs roughy 8% better than APGD-$\ell_1$ on Square Attack-$\ell_\infty$, though $\ell_1$ specific robustness is lower as seen with the Square-$\ell_1$ attack.

We further verify that such black-box adversaries are indeed weaker than the suite of white-box attacks presented in the main paper, thereby helping confirm the absence of obfuscated gradients in the proposed NCAT trained model.

## 2 Generalization to Unseen Domains

In the right-hand partition of Table-1, we present evaluations of the NCAT-$\ell_1$ and NCAT trained models on domain shifts that are not seen during training. We observe that the single-step trained models generalize well to images with common corruptions, obtaining 67% and 65% on the CIFAR10-

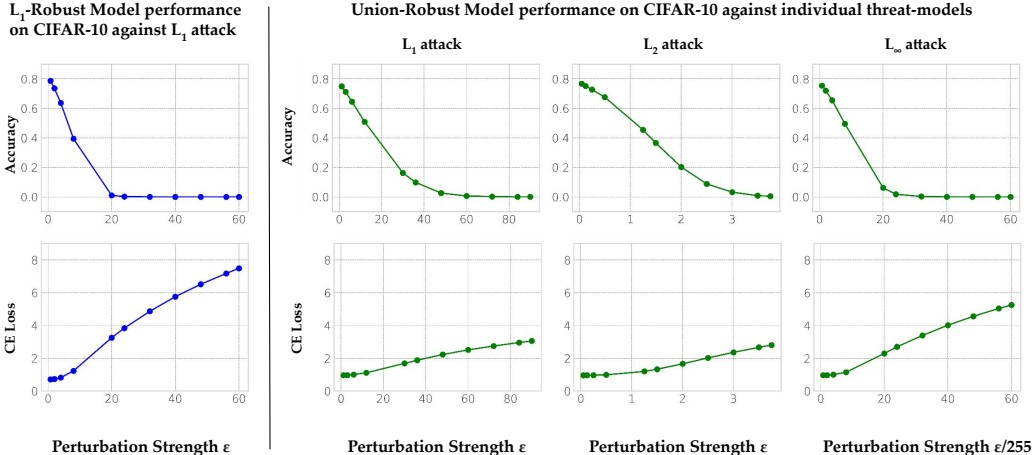

Figure 1: **Robustness across varying Perturbation Strengths** Row-1: Robust Accuracy is plotted for APGD-CE adversaries of different perturbation strengths for the NCAT-$\ell_1$ model in the left partition, and for the NCAT model robust in the right partition. Row-2: Cross-Entropy Loss is plotted for APGD-CE adversaries of different perturbation strengths for the NCAT-$\ell_1$ model in the left partition, and for the NCAT model robust in the right partition.

C dataset [Hendrycks and Dietterich, 2019] with the highest severity setting (5). The slight increase (0.2%) in the case of the NCAT-$\ell_1$ is likely due to the base clean accuracy being higher as compared to the NCAT model. Similarly, the APGD-$\ell_1$ trained model obtains higher accuracy on CIFAR10-C largely due to higher performance on clean samples. We also evaluate the model on Elastic and Gabor Transformations as introduced by Kang et al. [2019]. For Elastic image distortions, the NCAT model performs significantly better (+17.4%) as compared to the NCAT-$\ell_1$ which was trained solely against $\ell_1$ adversaries. Further, we observe that single-step training with NCAT or NCAT-$\ell_1$ achieves higher accuracy as compared to the APGD-$\ell_1$ trained model, with even NCAT-$\ell_1$ achieving an improvement of 5.4% on Elastic distortions over the latter, indicating the improved generalization seen with single-step training. However, for other distortions such as Gabor, prediction accuracy is significantly lower for all three models.

## 3   Gradient Masking Checks and Adaptive Attacks

In order to verify that the white-box attacks utilized are indeed effective in identifying strong adversaries within the considered threat model of interest, we present more detailed robust evaluations [Athalye et al., 2018] for the proposed NCAT trained ResNet-18 model in Fig.-1. Here, we present the accuracy versus epsilon plot, and cross-entropy loss versus epsilon plot for the NCAT-$\ell_1$ model in the first column on $\ell_1$ APGD-CE [Croce and Hein, 2020] adversaries. In the latter three columns, we present the same metrics on $\ell_1$, $\ell_2$ and $\ell_\infty$ APGD-CE attacks for various values of epsilon for the NCAT model trained to be robust against the union of such adversaries. In each case, we observe that the robust accuracy monotonically decreases to zero as the perturbation budget ($\varepsilon$) is increased. Further, the cross-entropy loss monotonically increases as the perturbation budget ($\varepsilon$) is increased. This shows that gradient-based white-box attacks are strong and effective, with a smooth local loss landscape, indicating the absence of gradient masking in the single-step defenses NCAT-$\ell_1$ and NCAT.

Further, we evaluate the NCAT defense against adaptive adversaries that incorporate modified objectives to obtain stronger attacks, since we assume that adversaries are cognizant of the training methodology used. We thus maximise the Nuclear Norm objective, (Eq-1 of the Main paper) to generate adaptive adversaries:

$$\widetilde{L} = \ell_{CE}\left(f_\theta(X+\Delta), Y\right) + \lambda \cdot ||f_\theta(X+\Delta) - f_\theta(X)||_*$$ (1)

Since the AutoAttack framework utilizes automatic updates to the step-size with restarts at the iteration that maximizes the overall loss, the incorporation of the Nuclear norm regularizer is sub-

optimal since batch-statistics across different images weaken the attack due to the reduced specificity in perturbations. We further implement an $\ell_1$-version of GAMA-PGD [Sriramanan et al., 2020] to incorporate the Nuclear norm objective, with a decaying coefficient for the regularization term in order to mitigate this effect. However, we find that this adaptive adversary is weak once again, with NCAT achieving 75.3% accuracy. Thus, we find that the adaptive attacks are not stronger than the evaluations performed using AutoAttack as presented in the Main paper, and that the latter is sufficient to obtain a reliable estimate of the worst-case $\ell_1$ accuracy obtained by the NCAT model.

# 4 Implementation Details and Training Methodology

## 4.1 Details on Datasets

In this work, we present our evaluations on the CIFAR-10 [Krizhevsky et al., 2009] and ImageNet-100 [Russakovsky et al., 2014] datasets, as they have come to form the benchmark datasets for robust evaluations.

CIFAR-10 [Krizhevsky et al., 2009] is a ten-class dataset, consisting of $32 \times 32$ sized RGB images arising from the following categories: "airplane", "automobile", "bird", "cat", "deer", "dog", "frog", "horse", "ship" and "truck". The test set of CIFAR-10 consists of 10,000 images, and the original training set consists of 50,000 images. The latter is split in practice, to form 49,000 training images and a hold-out validation set of 1000 images. On this dataset, we present robust evaluations against adversaries constrained under an $\ell_1$ ball of radius 12, $\ell_2$ ball of radius 0.5 and an $\ell_\infty$ ball of radius 8/255, similar to the setting considered in prior work [Maini et al., 2020, Croce and Hein, 2020].

ImageNet-100 is a hundred-class subset of the original ImageNet Large Scale Visual Recognition Challenge [Russakovsky et al., 2014], wherein every tenth class by WordNet ID order is retained, similar to the methodology followed by Laidlaw et al. [2021]. This dataset consists of $224 \times 224$ sized RGB images arising from a diverse set of classes. Given the high-dimensional nature of the images, and the diversity of classes, it is quite challenging to train robust models effectively on this dataset. On this dataset, we present robust evaluations against adversaries constrained under an $\ell_1$ ball of radius 255, $\ell_2$ ball of radius 1200/255 and an $\ell_\infty$ ball of radius 4/255, similar to Laidlaw et al. [2021]. Furthermore, images in this dataset are more realistic, with higher visual fidelity as compared to CIFAR-10. We thus present results on the unseen Neural Perceptual Threat Model (NPTM) [Laidlaw et al., 2021] on this dataset in Table-3 of the Main Paper using the Perceptual Projected Gradient Descent (PPGD) attack for the medium NPTM bound (0.5).

## 4.2 Training and Hyperparameter Details

In this work, all training and experimental evaluations were performed using Pytorch [Paszke et al., 2019]. We primarily utilize the ResNet-18 [He et al., 2016] architecture for both the CIFAR-10 and ImageNet-100 datasets. In addition, we present results on models trained on CIFAR-10 using the WideResNet-28-10 [Zagoruyko and Komodakis, 2016] architecture, that is, a WideResNet network with a depth of 28, and a width-factor of 10. We utilise a 100-epoch training schedule for the ResNet-18 models, and a 50-epoch regime for training WideResNet models. In all training runs, we use a cyclic schedule [Smith, 2015], with the maximum learning rate set to 0.1. We further utilize the Stochastic Gradient Descent (SGD) optimizer using a momentum parameter set to 0.9 and weight-decay of 5e-4. Further, we utilize Random-Crop and Random-Horizontal-Flip as augmentations for training images. Similar to prior works [Izmailov et al., 2018, Sriramanan et al., 2021], we utilize Stochastic Weight Averaging with the exponential parameter $\tau$ being largely optimal, together with a setting of 0.9998 with a batch-size of 64. With the introduction of random noise as initialization for the adversarial attack, $\ell_1$-based projections are need with pixel constraints. For this, we utilize an implementation by Croce and Hein [2021], together with linear scaling (=10) of the gradient in order to balance the relative scale to random noise. For NCAT-$\ell_1$, we set the coefficient of the Nuclear Norm regularizer $\lambda$ to 5, and for NCAT we use $\lambda = 3$ for $\ell_1$ adversaries and $\lambda = 5$ for $\ell_\infty$ adversaries to achieve robustness against the union of threat models. The proposed approach NCAT requires the same computational complexity in training as Nuclear Norm Adversarial Training (NuAT), and thus achieve the same reduction in computational requirements over existing multi-step approaches as reported by Sriramanan et al. [2021]. We use Nvidia RTX 2080 TI and Nvidia RTX A4000 GPU cards for training and experimental evaluations.

## 4.3 Details on Curriculum Schedule

As explained in Section-4.1 of the Main Paper, we propose to utilize a curriculum schedule for training on adversarial perturbations of increasing difficulty over the training regime. To do so, we linearly increase the radius of the $\ell_p$ ball considered for generating adversaries, thereby significantly reducing the extent of overfitting and eliminating catastrophic failure entirely during training. Further, we linearly increase the coefficient of the Nuclear norm regularization term $\lambda$ in-sync with the increase in $\ell_p$ radii. These techniques are particularly efficacious when we seek to achieve robustness against multiple threat models simultaneously, since different threat models can offer relatively different strengths of adversaries as the radii are increased during training. Similar to Sriramanan et al. [2020], we also set the value of $\lambda$ used in the attack to zero in alternate minibatches, in order to further boost diversity of adversaries seen during the training regime. In practice, we require that the model is trained on the final union of threat models for a sufficiently short duration, comprising of a few epochs of training. Hence, we linearly ramp up the $\ell_p$ radii such that adversaries are generated from the final threat model of interest in the last 10 epochs of training, following which the radii are kept constant.

## 5 Ablation Analysis

Table 2: **Ablations on CIFAR-10:** Prediction accuracy (%) of ResNet-18 models trained on the $\ell_1$ threat model using NCAT-$\ell_1$ (left), and on the union of $\ell_1$, $\ell_2$ and $\ell_\infty$ threat models using NCAT (right). Robust accuracy is computed using only $\ell_1$ adversaries in the left partition, while worst-case accuracy over adversaries constrained under the union of $\ell_1$, $\ell_2$ and $\ell_\infty$ threat models is presented in the right partition.

| Method | Clean Acc | $\ell_1$ Robust Acc | Method | Clean Acc | Worst-Case Acc |
|---|---|---|---|---|---|
| A1: RFGSM-$\ell_1$ | 89.9 | 0.0 | A5: Exp. Wts. Samp | 79.9 | 39.2 |
| A2: RFGSM-$\ell_1$ + Early-stop. | 71.8 | 32.5 | A6: NCAT-AVG | 79.1 | 40.4 |
| A3: NuAT-$\ell_1$ | 92.8 | 1.2 | A7: NCAT $p = 0.4$ | 80.9 | 42.4 |
| A4: NuAT-$\ell_1$ + Early-stop. | 81.2 | 36.1 | A8: NCAT $p = 0.6$ | 80.1 | 42.0 |
| NCAT-$\ell_1$ | 80.6 | 55.5 | NCAT | 80.5 | 42.5 |

In this section, we perform ablative experiments to study the significance of different components in the proposed defense. In the left partition of Table-2, we present results for various $\ell_1$ trained models, while the right partition corresponds to models that are trained to be robust against adversaries under the union of $\ell_1$, $\ell_2$ and $\ell_\infty$ threat models. In Ablations A1 and A2, we present results obtained using RFGSM training [Wong et al., 2020], wherein we note that catastrophic failure occurs early during the course of training. Even with early-stopping as suggested by Wong et al. [2020], the model obtains low clean accuracy (71.8%), and subpar robust accuracy due to the early collapse in training. We observe a similar phenomenon with Nuclear Norm adversarial training (A3,A4), wherein the model undergoes failure at a delayed phase as compared to RFGSM trained models. Thus, though NuAT obtains improved results, catastrophic failure during training results in the sub-par models with very low robust performance (36.1%). However, with the curriculum schedule as explained in Section-4.3 and Section-4.1 of the Main Paper, the training dynamics in NCAT is highly stabilized, resulting in the first $\ell_1$ robust model trained solely using single-step adversaries.

In the right partition, we first present ablation A5, wherein the frequency of sampling adversaries from different threat models is dynamically altered according to an exponential weights algorithm as proposed by Awasthi et al. [2021], based on metrics recorded on a hold-out validation set. In practice, these updates are seen to be excessively sensitive to the degree of convergence achieved by adversaries on the validation set resulting in lower robust accuracy on the union of adversaries (39.2%), and further requires additional hyperparameter tuning for the exponential weighting, along with an added computational budget for recording validation performance at each epoch. In ablation A6, we present NCAT-AVG wherein a single-step adversary is generated for each threat model separately, and thus has a 3x computational overhead as compared to the base NCAT defense. Further, we observe that the robust accuracy under the union of threat models is reduced despite the increase in training cost, and is accompanied with reduction in clean performance as well. Lastly, we present ablations A7 and A8

Table 3: **Stability across Reruns** Prediction accuracy (%) of ResNet-18 models trained on the $\ell_1$ threat model using NCAT-$\ell_1$ (left), and on the union of $\ell_1$, $\ell_2$ and $\ell_\infty$ threat models using NCAT (right). Robust accuracy is computed using only $\ell_1$ adversaries in the left partition, while worst-case accuracy over adversaries constrained under the union of $\ell_1$, $\ell_2$ and $\ell_\infty$ threat models is presented in the right partition.

| NCAT-$\ell_1$ | Clean Acc | $\ell_1$ Robust Acc | NCAT | Clean Acc | Worst-Case Acc |
|---|---|---|---|---|---|
| Rerun-1 | 80.71 | 55.60 | Rerun-1 | 80.46 | 42.58 |
| Rerun-2 | 80.43 | 55.67 | Rerun-2 | 80.52 | 42.51 |
| Rerun-3 | 80.56 | 55.32 | Rerun-3 | 80.38 | 42.45 |
| Rerun-4 | 80.39 | 55.43 | Rerun-4 | 80.56 | 42.27 |
| Rerun-5 | 80.60 | 55.51 | Rerun-5 | 80.47 | 42.46 |
| Mean | 80.54 | 55.51 | Mean | 80.48 | 42.45 |
| Std-Dev | 0.13 | 0.14 | Std-Dev | 0.07 | 0.11 |

where the frequency of sampling $\ell_\infty$ based adversaries is changed to $p = 0.4, p = 0.6$ respectively. In practice, it is highly plausible that a subset of specified threat models is significantly simpler to achieve robustness as compared to other adversaries. This sampling mechanism helps incorporate the same in a simple manner, and subsumes NCAT which utilizes $p = 0.5$ for all experiments. This sampling technique helps provide yet another mechanism for trading off robustness for one threat model against another, as per design or specification requirements. For example, while both ablation models A7, A8 achieve similar $\ell_p$-union robustness (42.4% and 42%), on the specific $\ell_1$ and $\ell_\infty$ threat models, A7 achieves 48.8% and 44.7% robust accuracy respectively, while A8 achieves 45.1% and 46.5% robust accuracy respectively. This clearly indicates the trade-off achieved with sampling, wherein with $p = 0.6$, the model achieves higher $\ell_\infty$ robustness, alongside a reduction in $\ell_1$ accuracy.

## 6 Stability of NCAT

In Table-3, we analyze the variation in prediction accuracy for both clean and adversarial samples, for ResNet-18 models trained on CIFAR-10 using five different random seeds on the same Nvidia RTX 2080 TI GPU, with hyperparameter frozen across reruns. In the left-partition of the table, we present results for the model trained to be robust against $\ell_1$ adversaries in particular, using NCAT-$\ell_1$, while in the partition on the right, we present results for the model trained to be robust against adversaries under the union of $\ell_1$, $\ell_2$ and $\ell_\infty$ threat models. We observe that models trained using either NCAT-$\ell_1$ or NCAT are very stable across reruns, with variance levels similar to that reported from multi-step training approaches such as PGD-AT [Madry et al., 2018, Rice et al., 2020] and TRADES [Zhang et al., 2019]. Furthermore, we note that NCAT based adversarial training does not suffer from catastrophic failure during any of the runs, in sharp contrast to that seen from RFGSM or NuAT based training, wherein catastrophic failures are observed in almost every training run.

## 7 Loss curvature analysis of Adversarial Training with Curriculum

To further investigate why the proposed curriculum schedule is effective in assisting single-step training, it is imperative to analyze the local-linearity of the loss surface. To do so, we compute the curvature of the local cross-entropy loss surface across epochs during training, similar to that performed by Moosavi-Dezfooli et al. [2019], Ortiz-Jiménez et al. [2022]. During the course of robust training with a curriculum schedule, the model receives supervision to maintain smoothly varying function values for gradual, progressive increase in perturbation radii. As the model receives explicit supervision for intermediate radii with single-step adversaries following the curriculum, the degree of local oscillations is damped as training progresses. In sharp contrast, during catastrophic overfitting in single-step training, prior work Ortiz-Jiménez et al. [2022] identifies a coincident drastic increase in the local curvature, indicating the poor approximation achieved using single-step optimization alone. We hypothesize that both the curriculum schedule and the nuclear norm regularizer independently reduce local curvature. We thus plot the curvature of models across different epochs in Figure-2, wherein we note striking similarities with respect to the accuracy and loss curves presented in Figure-1

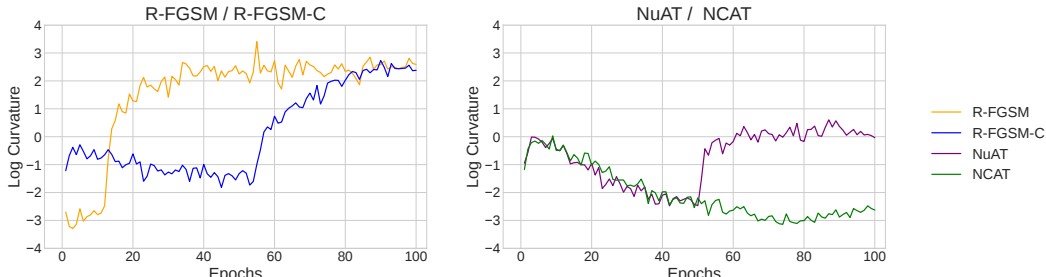

Figure 2: **Catastrophic Overfitting in $\ell_1$ Adversarial Training:** We also investigate the stability of curriculum based adversarial training through loss curvature analysis following Moosavi-Dezfooli et al. [2019], Ortiz-Jiménez et al. [2022]. We plot the log curvature of the Cross Entropy loss over the test-set of CIFAR-10 across different training epochs. On the left, we compare the log-curvature of R-FGSM with its curriculum based counterpart (R-FGSM-C), while on right we compare Nuclear Norm Adversarial Training (NuAT) with the proposed method, NCAT.

of the Main paper. We observe that for R-FGSM, the curvature of the loss grows by several orders of magnitude with the onset of catastrophic overfitting. We further observe that with the curriculum schedule RFGSM training is more stabilized, and that such onset of failure is delayed significantly. For training methods with nuclear norm, we initially observe a trend similar to that of RFGSM training with curriculum, till the onset of failure in NuAT around epoch 50. Thus while the curvature estimates proposed by Moosavi-Dezfooli et al. [2019] agree closely for both NuAT and NCAT for the initial epochs, the latter method is stable throughout the entire training regime. This indicates that while local curvature is an important tool to analyze training methods, low-curvature alone is not sufficient to guarantee stable training devoid of catastrophic overfitting.