# OpenReview forum: "Toward Efficient Robust Training against Union of $\ell_p$ Threat Models"
_NeurIPS.cc/2022/Conference — NeurIPS 2022 Accept_

### Official Review · Reviewer_mBSh · 2022-07-08

**Rating:** 7
**Confidence:** 5
**Soundness:** 4 excellent
**Presentation:** 3 good
**Contribution:** 3 good

**Summary:**

The paper proposes a method, NCAT, to obtain models robust wrt $\ell_1$ or multiple $\ell_p$-norms efficiently, that is with single step adversarial training. NCAT combines Nuclear-Norm Adversarial Training (NuAT) with a curriculum for the size of the perturbations seen at training time to prevent catastrophic overfitting in 1-step adversarial training. Further, the paper applies NCAT to the multiple norms scenario, achieving results close to those of more expensive multi-step methods.

**Questions:**

- While the paper is well-written, I think some more details about the curriculum (and maybe some training details) could belong to the main part, since that's one of the main components of NCAT. Also in Table 1, NCAT+ appears but I couldn't find where it's defined.

- An efficient method is particularly useful when training is particularly expensive, then it'd be interesting to scale NCAT to the full version of ImageNet.

**Limitations:**

The limitations are sufficiently addressed.

**Strengths And Weaknesses:**

Strengths
- The goal of reducing the cost of adversarial training without degrading (too much) the resulting robustness is well established, and to my knowledge this is the first work to extend single step adversarial training to $\ell_1$ and multiple norms.

- The proposed method is simple and effective in the experimental evaluation on both CIFAR-10 (with two architectures) and ImageNet-100. Several additional experiments are presented in the supplements to further analyze and support NCAT.

Weaknesses
- NCAT is built combining several existing techniques (NuAT, curriculum for $\epsilon$, steepest ascent direction for the threat models of $\ell_1$ with the image domain constraints, alternating $\ell_1$ and $\ell_\infty$ when training for multiple norms), which to some extent limits the contributions of the paper.

Overall, I think that the novelty of the method is somehow limited, showing how it's possible to achieve with single step adversarial training robustness in challenging threat models a meaningful contribution.

---

> ### Author Response · Authors · 2022-08-02
> **Response to Reviewer mBSh**
>
> We thank the reviewer for the detailed comments, and address the specific comments posted by the reviewer mBSh below:
>
> - NCAT$^+$: The relative trade-off between clean accuracy and robustness can be adjusted depending on individual use-case requirements by tuning the coefficient of the nuclear norm regularization term in the overall loss objective. This is presented in Table-1 as NCAT$^+$, wherein this coefficient is increased so as to achieve greater $\ell_1$ robustness, at the cost of lower clean accuracy. We shall also certainly move more details on the curriculum schedule to the main paper in the final version as suggested.
>
> $ $
>
> - Results on ImageNet-1K: Due to the exceedingly large memory and computational requirements for training even a non-robust (standard) model on ImageNet, in the paper we present results with adversarial training on a 100-class subset, identical to the one considered by Laidlaw et al. (2021) [1]. However, we certainly agree that it would be interesting to scale NCAT to the full version of ImageNet. We shall try our best to work toward this, and try to include results on the entire ImageNet dataset with 1000 classes in the final version of the paper.
>
> $ $
>
> We sincerely thank the reviewer for the support for acceptance. We greatly appreciate the valuable comments and suggestions, and we will certainly try to incorporate these in the final version of the paper.
>
> $ $
>
> [1] C. Laidlaw, S. Singla, and S. Feizi, Perceptual adversarial robustness: Defense against unseen threat models. In International Conference on Learning Representations (ICLR), 2021

---

### Official Review · Reviewer_4QVR · 2022-07-09

**Rating:** 8
**Confidence:** 4
**Soundness:** 4 excellent
**Presentation:** 4 excellent
**Contribution:** 3 good

**Summary:**

This work tackles the lucrative goal of multi-norm perturbation robustness, which is largely constrained by efficient robustness against $L_1$-norm adversaries in the current literature. The authors propose a technique inspired by the Nuclear-Norm regularizer that helps achieve robustness against a union of adversaries efficiently. The curriculum-based method has clear performance gains, especially with complex datasets and large models.

**Questions:**

There is a non-trivial drop in clean-model accuracy when using the proposed method. I believe this is probably a side effect of learning more robust models, thereby limiting weak dataset-specific signals that the model might otherwise be latching on to. Nonetheless, it would be nice if the authors could address this drop better and do some kind of qualitative analysis. For instance, visual/statistical inspection of the (additional) cases where the models go wrong may help identify problematic examples (mislabeled), or specific concepts that seem to be hard if aiming for robustness as well.

**Limitations:**

Although societal impact is addressed in Section 7, I do not see much discussion around limitations.

**Strengths And Weaknesses:**

# Strengths
- The paper is well written, with easy-to-follow reasoning for design/algorithmic choices, buildup, and review of relevant literature. Evaluation is also thorough (although I would prefer to see at last least some mention of deviation in results across runs).
- The observation (and perhaps affirmation) of $L_1$ and $L_{\infty}$ being the hardest threat models while aiming for union robustness is nice.

# Weaknesses
As such, I see no visible weaknesses in the paper, apart from minor comments mentioned below (and in the questions section). Good job!

# Minor comments
- Table 1: What does the $^+$ in NCAT$^+$ stand for? Please clarify in the table description.

---

> ### Author Response · Authors · 2022-08-02
> **Response to Reviewer 4QVR**
>
> We thank the reviewer for the detailed comments, and address the specific comments posted by the reviewer 4QVR below:
>
> - **Stability Across Reruns**
>
>     We analyze the variation in prediction accuracy for both clean and adversarial samples, for ResNet-18 models trained on CIFAR-10 using five different random seeds in Table-3 of the Supplementary material. We observe that the method is indeed stable across reruns, with a low standard deviation of 0.07 and 0.11 respectively for Clean accuracy and Worst-case Union accuracy for NCAT, and a standard deviation of 0.13 and 0.14 respectively for Clean accuracy and $\ell_1$ robust accuracy for NCAT-$\ell_1$.
> $ $
> - NCAT$^+$: The relative trade-off between clean accuracy and robustness can be adjusted depending on individual use-case requirements by tuning the coefficient of the nuclear norm regularization term in the overall loss objective. This is presented in Table-1 as NCAT$^+$, wherein this coefficient is increased so as to achieve greater $\ell_1$ robustness, at the cost of lower clean accuracy.
>
> $ $
>
> - **Qualitative Analysis of Misclassified Images**
>
>     - As mentioned in the review, we too believe that the drop in clean accuracy during robust training can be attributed to the learning of different sets of features. Tsipras et al. [1] demonstrate this in greater detail, wherein standard models are shown to utilize any feature that is weakly correlated with the ground-truth label, while the min-max optimization routine in adversarial training forces robust models to only utilize strongly-correlated features that cannot become anti-correlated with the label under imperceptible adversarial perturbations.
>
>     - To better understand the relation between different threat models, we did perform a visual inspection of test images that were correctly classified under an $\ell_{\infty}$ attack, but misclassified under an $\ell_1$ attack and vice-versa on the CIFAR-10 dataset. However, this did not lead to any concrete conclusions, as we were unable to identify any significant, conspicuous trends. We also did not observe problematic or mislabelled samples in this setting.
>
>     - We also visually inspected images that were misclassified by an NCAT model without any adversarial perturbation added, yet correctly classified by a normally trained model. Here too, no significant trend was largely apparent across different images. Interestingly, a small number of clean test images were misclassified by the normally trained model, yet correctly classified by the robust model. In this setting, we did observe some test images that were relatively more challenging with a greater extent of ambiguity. In general, we also do note that images from animal-based classes such as Bird, Cat, Dog and Deer of CIFAR-10 are misclassified relatively more frequently, since they often contain more visually challenging image samples. We shall certainly try to analyze this skew in performance with quantitative tools in the final version of the paper.
>
> $ $
>
> We sincerely thank the reviewer for the support for acceptance. We greatly appreciate the valuable comments and suggestions, and we will certainly incorporate them in the final version of the paper.
>
> $ $
>
> [1] D. Tsipras, S. Santurkar, L. Engstrom, A. Turner and A. Madry, Robustness May Be at Odds with Accuracy. In International Conference on Learning Representations (ICLR), 2019

---

> > ### Comment · Reviewer_4QVR · 2022-08-08
> > **Issues Addressed**
> >
> > Dear authors,
> >
> > Thanks for the rebuttal. I am satisfied with the response (and the paper in its current form, with the proposed changes), and am changing my rating to reflect this. Good luck :)

---

### Official Review · Reviewer_LkCw · 2022-07-12

**Rating:** 4
**Confidence:** 5
**Soundness:** 2 fair
**Presentation:** 3 good
**Contribution:** 2 fair

**Summary:**

The paper proposes an efficient robust training method to achieve adversarial robustness on multiple threat models (l_1,l_2,l_\inf). Specifically, it first study the adversarial training on  l_1 norm where normal adversarial training method could achieve a good performance. By combining the previously proposed nuclear-norm adversarial training and curriculum schedule, it finds it could achieve a better robust accuracy against l_1 adversarial attack. As l_2 and l_inf threat model have a similar behavior on the robustness, the paper then choose to alternatively switch the threat model to do nuclear-norm adversarial training to improve the efficiency. The experiments are conducted on cifar10 and imagenet100 on ResNet18 and wideresnet28. It shows the proposed method could achieve a better worst-case robustness while maintain a good scalabliilty.

**Questions:**

1. Does the weight average on the model parameter matter in Algorithm 1 line 13? I don't see any explanation on that. Also, which parameter is used in the end? theta or w?
2. The curriculum scheduling part is not clear to me. Could you elaborate more on that?
3. If the threat model doesn't include l_1 threat model, will the proposed method still have a good performance?

**Limitations:**

The papers has discussed its limitation on provable certificates on different l_p threat models and their union. However, I think one of big limitation is the paper requires the union to have l_1 threat model.

**Strengths And Weaknesses:**

Pros:
1. The paper is in general well-written and easy to follow except for the curriculum scheduling part.
2. The experiment result shows the proposed method could achieve a worst-case robust accuracy.

Cons:
1. The novelty of proposed method is quite limited. It is a fairy combination on nuclear norm adversarial training with the curriculum schedule. And curriculum schedule is also discussed in the previous works such as [1]. It is still unclear why the curriculum schedule is useful in the adversarial training or the proposed schedule only works for the l_1 threat model.
2. The main claim of improving the union of threat model is actually improving the adversarial training's performance on the l_1 threat model, which is not equal to improve the performance over the union. In other words, if the union doesn't include l_1 threat model, will the proposed method still outperform the baseline?
3. Although the proposed method is better in terms of the worst-case acc, they are sometimes worse on the average. The main improvement I think is on the training efficiency.

Minor:
1. Typos: line 219 a_\sigma(1)\geq a_\simga(1)..
2. The dash line in Figure 1 (b)(c) is inconsistent with the legend.

[1] Zhang, Jingfeng, et al. "Attacks which do not kill training make adversarial learning stronger." International conference on machine learning. PMLR, 2020.

---

> ### Author Response · Authors · 2022-08-02
> **Response to Reviewer LkCw (Part-1)**
>
> We thank the reviewer for the detailed comments, and address the specific comments posted by the reviewer LkCw below:
>
> - **Curriculum Schedule**
>
>
>     - In order to improve the efficacy of single-step adversarial training, we propose to utilize a dynamically varying perturbation budget during training, to ensure that the model receives supervision on increasingly difficult adversaries as determined by a Curriculum schedule. Different curricula are seen to be effective in practice, including those that incorporate the robust accuracy on training samples observed at a given epoch to dynamically vary the perturbation budget at different rates. However, these complex curricula require additional fine-tuning of hyperparameters, and are sensitive to the $\ell_p$ threat model considered. We thus utilize a simple, linear curriculum wherein the perturbation size is increased from zero to $\varepsilon$, and kept constant for the last few epochs (10 for NCAT), to facilitate adequate fine-tuned training on the final threat model of interest.
>
>     - The curriculum schedule proposed is particularly effective for single-step training given that it does not require sampling of adversarial statistics at different radii. In contrast, Zhang et al. [1] propose Friendly Adversarial Training (FAT) wherein early stopping is performed during the generation of the multi-step PGD attack. The authors propose to stop updating the perturbation as soon as the PGD attack successfully induces misclassification on the training image. This however inherently requires a multi-step adversarial generation approach to sample different stopping points for its implementation in practice.
>
>
>     - We would also like to highlight again, the relative difficulty in achieving robustness against $\ell_1$ attacks that are highly sparse in practice. Indeed, prior works note that even adversarial training using expensive 10-step $\ell_1$ adversaries generated from SLIDE is prone to catastrophic overfitting. While such phenomena are generally seen in single-step methods, the occurrence of such failure modes even with 10-step adversaries indicates the difficulty in identifying techniques to produce $\ell_1$ robust networks, while limiting the available budget to include efficient single-step adversaries alone.
>
> $ $
>
> - **Performance under other Threat Models**
>
>     Furthermore, the curriculum schedule is effective on threat models other than $\ell_1$ as well, such as $\ell_{\infty}$ based training. While Nuclear Norm AT (Sriramanan et al., 2021 [2]) and RFGSM training with Early Stopping (Wong et al., 2020 [3]) achieves non-trivial robustness against $\ell_{\infty}$ based adversaries with a perturbation radius of $\varepsilon = 8/255$, the stability is decreased at larger radii such as $\varepsilon = 10/255$. The proposed method with curriculum based training is seen to significantly improve the robustness achieved using the same budget of a single-step attack per minibatch, as shown in the table below. Robust evaluations are presented for an $ \ell_{\infty}$ radius of $\varepsilon = 10/255$ and $ \ell_{2}$ radius of $\varepsilon = 0.5$, though each model is trained explicitly only on the $\ell_{\infty}$ adversaries. We observe that the curriculum schedule is effective in enhancing RFGSM training with Early Stopping, improving both clean and robust accuracy. Further, NuAT is seen to undergo catastrophic failure at this radius and thus requires early-stopping to identify a robust model, while NCAT is stable throughout training, and is thus seen to achieve significantly higher clean and robust accuracy.
>
>
>
>  | Method                          | Clean Acc. | Linf Acc. | L2 Acc. | Worst Acc. | Average Acc. |
> |---------------------------------|:----------:|:---------:|:-------:|:----------:|:------------:|
> | RFGSM+Early Stopping            |    68.2    |    28.9   |   49.0  |    28.9    |     38.9     |
> | RFGSM+Early Stopping+Curriculum |    83.8    |    31.9   |   57.8  |    31.8    |     44.9     |
> | NuAT+Early Stopping             |    74.2    |    32.8   |   53.4  |    32.8    |     43.1     |
> | NCAT - $\ell_{\infty}$                            |    80.6    |    41.3   |   60.5  |    41.3    |     50.9     |

---

> > ### Author Response · Authors · 2022-08-02
> > **Response to Reviewer LkCw (Part-2)**
> >
> > - **Loss Curvature analysis with Curriculum Training**
> >
> >     To further investigate why the proposed curriculum schedule is effective in assisting single-step training, it is imperative to analyze the local-linearity of the loss surface. To do so, we compute the curvature of the local cross-entropy loss surface across epochs during training, similar to that performed by [4,5]. During the course of robust training with a curriculum schedule, the model receives supervision to maintain smoothly varying function values for gradual, progressive increase in perturbation radii. As the model receives explicit supervision for intermediate radii with single-step adversaries following the curriculum, the degree of local oscillations is damped as training progresses. In sharp contrast, during catastrophic overfitting in single-step training, prior work [4] identifies a coincident drastic increase in the local curvature, indicating the poor approximation achieved using single-step optimization alone. We hypothesize that both the curriculum schedule and the nuclear norm regularizer independently reduce local curvature. We thus plot the curvature of models across different epochs in Figure-2 of the revised Supplementary document, wherein we note striking similarities with respect to the accuracy and loss curves presented in Figure-1 of the Main paper. We do note however that while local curvature can be an important tool to analyze training methods, low-curvature alone is not sufficient to guarantee stable training devoid of catastrophic overfitting.
> >
> > $ $
> >
> > - Weight Averaging: We utilize Stochastic Weight Averaging (SWA) since it is known to produce flatter loss basins and leads to better robust generalization. Thus, while the main model (denoted with parameters $\theta$ in Algorithm-1) is also robust, the final NCAT models used for evaluations are the exponentially averaged model weights, denoted with parameters $\omega$ in Algorithm-1.
> >
> > $ $
> >
> > - Average Case Performance: We sincerely apologize for a typo wherein the average-case accuracy for NCAT-$\ell_1$ in Table-1 was listed as 34.4%, instead of the correct value which was significantly higher at 53.6%. In general, we find that the proposed method improves upon both worst-case and average-case accuracy.
> >
> > - Typos: We thank the reviewer for pointing out these, we have fixed such typos in the rebuttal revision.
> >
> > $ $
> >
> > We thank the reviewer again for their valuable comments and suggestions. We kindly ask if the reviewer would consider increasing their score to support acceptance of the paper if their concerns or questions have been addressed. We would also be glad to engage further during the author-reviewer discussion period.
> >
> > $ $
> >
> > [1] J. Zhang, X. Xu, B. Han, G. Niu, L. Cui, M. Sugiyama and M. Kankanhalli, Attacks Which Do Not Kill Training Make Adversarial Learning Stronger. In International Conference on Machine Learning (ICML), 2020
> >
> > [2] G. Sriramanan, S. Addepalli, A. Baburaj, and V. B. Radhakrishnan. Towards efficient and effective adversarial training. In Advances in Neural Information Processing Systems (NeurIPS), 2021
> >
> > [3] E. Wong, L. Rice and J. Z. Kolter, Fast is better than free: Revisiting adversarial training. In International Conference on Learning Representations (ICLR), 2020
> >
> > [4] G. Ortiz-Jiménez, P. Jorge, A. Sanyal, A. Bibi, P. Dokania, P. Frossard, G. Rogez and P. Torr, Catastrophic overfitting is a bug but also a feature, 2022
> >
> > [5] S. Moosavi-Dezfooli, J. Uesato, A. Fawzi and P. Frossard, Robustness via curvature regularization, and vice versa. In Conference on Computer Vision and Pattern Recognition (CVPR), 2019

---

> > ### Comment · Reviewer_LkCw · 2022-08-08
> > **Response to rebuttal**
> >
> > Thank author for the detailed rebuttal. Although some concerns have been cleared, I find the proposed ensemble relies way too much on the l_1 attack. At the same time, there is incremental difference in terms of novelty between the proposed methods with [1] which basically focus on the same setting and aim for improving the ensemble by l1 attack. Therefore, I would think the main contribution is actually to propose some trick to find how to do a quickly (one-step) PGD attack to defense over l_1 norm attack.
> >
> > [1] F. Croce and M. Hein. Adversarial robustness against multiple lp-threat models at the price of one and how to quickly fine-tune robust models to another threat model

---

> > > ### Author Response · Authors · 2022-08-09
> > > **Response to Reviewer LkCw (Part-3)**
> > >
> > > We thank the reviewer again for their response. We address the specific comments below:
> > >
> > > - **Efficacy on Threat Models other than $\ell_1$**
> > >
> > >     As discussed in the rebuttal above, NCAT is seen to be highly effective even in cases without the inclusion of the $\ell_1$ norm. With $\ell_{\infty}$ based training for example, we observe gains as large as 8%-12% in $\ell_{\infty}$ robust accuracy, and 6%-12% in clean accuracy over prior baselines as shown in the table above (Part-1 of rebuttal).
> > >
> > > $ $
> > >
> > > - **Crucial Need for $\ell_1$-based Analysis**
> > >
> > >     - While the $\ell_{\infty}$ threat model is the most well-studied, such $\ell_{\infty}$ trained models still offer very poor robustness to other equivalently valid imperceptible perturbations such as $\ell_1$ or PPGD attacks. Further, any work that seeks to achieve worst-case robustness in these generalized settings must necessarily improve upon individual performance as well. Given that the $\ell_1$ threat model has formed the primary bottleneck in achieving the same upto now, this important setting is a crucial area of focus in our paper as well.
> > >
> > >     - We would like to highlight that prior to this work, it was highly unclear and not known if robustness using efficient methods such as single-step training could even be achieved in the $\ell_1$ threat model, given that catastrophic failure was observed even with some multi-step training approaches.
> > >
> > > $ $
> > >
> > > - **Efficient Robust Training for Multiple Norms**
> > >
> > >     The need to identify efficient robust training methods is further accentuated when we require models to be robust against multiple threat models simultaneously, given the large computational overhead involved with multi-step adversarial training like PGD-AT.  Towards this, the proposed method NCAT continues to utilize only a single-step attack per minibatch to achieve worst-case robustness against the union of $\ell_p$ threat models.
> > >
> > >  $ $
> > >
> > > - **Contrast wrt EAT/Fine-Tuning [1]**
> > >
> > >     Though Croce and Hein [1] do indeed focus on the same setting of achieving robustness under the union of $\ell_p$ norm threat models, they propose a fundamentally different technique, wherein multi-step adversarial fine-tuning is performed on models that are already robust against an individual $\ell_p$ threat model. Thus, the proposed method NCAT forms an orthogonal finding, demonstrating efficient single-step training against $\ell_1$ and multiple norm adversaries from random initialization, without relying upon robust pre-trained models.
> > >
> > > $ $
> > >
> > >
> > > [1] F. Croce and M. Hein. Adversarial robustness against multiple lp-threat models at the price of one and how to quickly fine-tune robust models to another threat model

---

> ### Author Response · Authors · 2022-08-08
> **Author-Reviewer Discussion Period**
>
> Dear Reviewer LkCw, since the deadline for the author-reviewer discussion period is approaching soon, we kindly ask if you have any further questions. We would be glad to engage further during this discussion period.
>
> We kindly ask if the reviewer would consider increasing their score to support acceptance of the paper if their concerns or questions have been addressed adequately.

---

### Official Review · Reviewer_TwcU · 2022-07-30

**Rating:** 4
**Confidence:** 3
**Soundness:** 2 fair
**Presentation:** 3 good
**Contribution:** 2 fair

**Summary:**

This paper presents a robust training against the union of $l_p$ threat models, i.e., training a robust model in l_inf, l_2, l_1, threat models. The key idea is to use a nuclear norm-based training procedure called NCAT.

**Questions:**

Please report the actual computation time for training using NCAT vs. other baseline adversarial training methods.

**Limitations:**

yes

**Strengths And Weaknesses:**

Strengths
-----------
- NCAT can achieve SOTA robust accuracy while requiring a single-step attack budget per mini-batch
- NCAT can scale to large datasets


Weaknesses
--------------
- The actual training performance (in terms of computation time) is not reported. Therefore, it is hard to compare the actual performance of NCAT against other robust training methods.

- Prior work (Sriramanan et al.) 2021 has already used nuclear norms in the context of adversarial training, albeit for single-step attacks.

---

> ### Author Response · Authors · 2022-08-02
> **Response to Reviewer TwcU (Part-1)**
>
> We thank the reviewer for the detailed comments, and address the specific comments posted by the reviewer TwcU below:
>
> - **Computation Time**
>
>     As requested, we provide computation training times in the table below. For these experiments, we present the time required for adversarial training on the CIFAR-10 dataset, as recorded on an NVIDIA RTX A4000 GPU card. We thus observe that the proposed approach NCAT/NCAT-$\ell_1$ trains significantly faster as compared to multi-step training methods. In the first row, we include the training time for RFGSM-$\ell_1$ with a near-identical codebase for reference, though the method itself fails to produce adequate robustness against $\ell_1$ adversaries. We also remark that the $\ell_1$ projection step is more computationally involved than clipping as in $\ell_{\infty}$ projections. We also note that in practice, the training can be further accelerated using mixed-precision arithmetic [1], or by efficiently pre-processing and caching datasets [2] since data-loading can often create significant bottlenecks in the overall computation pipeline.
>
>
> | Method   | Architecture | Time/Epoch (sec) | Relative Speedup |
> |----------|:--------------:|:----------------:|:----------------:|
> | RFGSM-$\ell_1$ | RN-18        |        62        |       0.75      |
> | APGD-$\ell_1$  | RN-18        |        293       |       3.53      |
> | NCAT-$\ell_1$  | RN-18        |        85        |       1.02      |
> | AVG      | RN-18        |        726       |       8.74      |
> | MAX      | RN-18        |        705       |       8.49      |
> | NCAT     | RN-18        |        83        |       1.00      |
> | | | | |
> | APGD-$\ell_1$  | WRN-28-10    |       1219       |       3.39      |
> | NCAT-$\ell_1$  | WRN-28-10    |        364       |       1.01      |
> | AVG      | WRN-28-10    |       3572       |       9.92      |
> | MAX      | WRN-28-10    |       3198       |       8.88      |
> | NCAT     | WRN-28-10    |        360       |       1.00      |
>
>
> $ $
>
> - **Stability Improvements**
>
>     While Nuclear Norm AT (Sriramanan et al., 2021 [3]) and RFGSM training with Early Stopping (Wong et al., 2020 [1]) achieves non-trivial robustness against $\ell_{\infty}$ based adversaries with a perturbation radius of $\varepsilon = 8/255$, the stability is decreased at larger radii such as $\varepsilon = 10/255$. The proposed method with curriculum based training is seen to significantly improve the robustness achieved using the same budget of a single-step attack per minibatch, as shown in the table below. Robust evaluations are presented for an $ \ell_{\infty}$ radius of $\varepsilon = 10/255$ and $ \ell_{2}$ radius of $\varepsilon = 0.5$, though each model is trained explicitly only on the $\ell_{\infty}$ adversaries. We observe that the curriculum schedule is extremely effective in improving stability of RFGSM training with Early Stopping, improving both clean and robust accuracy. Further, NuAT is seen to undergo catastrophic failure at this radius and thus requires early-stopping to find a robust model, while NCAT is stable throughout training, and is thus seen to achieve higher clean and robust accuracy.
>
>
>  | Method                          | Clean Acc. | Linf Acc. | L2 Acc. | Worst Acc. | Average Acc. |
> |---------------------------------|:----------:|:---------:|:-------:|:----------:|:------------:|
> | RFGSM+Early Stopping            |    68.2    |    28.9   |   49.0  |    28.9    |     38.9     |
> | RFGSM+Early Stopping+Curriculum |    83.8    |    31.9   |   57.8  |    31.8    |     44.9     |
> | NuAT+Early Stopping             |    74.2    |    32.8   |   53.4  |    32.8    |     43.1     |
> | NCAT - $\ell_{\infty}$                            |    80.6    |    41.3   |   60.5  |    41.3    |     50.9     |

---

> > ### Author Response · Authors · 2022-08-02
> > **Response to Reviewer TwcU (Part-2)**
> >
> > - **Contributions**
> >
> >     - We would also like to highlight the main contributions of the paper once again. First, the paper tackles the highly challenging setting of achieving robustness against $\ell_1$ constrained adversaries. It is particularly noteworthy that prior works point out the intricacies involved with $\ell_1$ adversarial training, in that even multi-step training methods (eg. SLIDE) can exhibit catastrophic failure. Thus, ensuring stable training with single-step adversaries alone is a significant challenge for $\ell_1$ robustness, and is successfully addressed in an efficient manner using the proposed approach, NCAT-$\ell_1$.
> >
> >     - Furthermore, the method extends in a simple yet effective manner to achieve robustness against adversaries under the union of $\ell_1,\ell_2$ and $\ell_{\infty}$ norms while again using the same budget of a single-step attack per minibatch. This is in sharp contrast to existing approaches that either utilize adversarial training methods which require multi-step attacks from different threat models that are computationally expensive in practice, or rely upon fine-tuning of pre-trained models that are robust with respect to a single-threat model.
> >
> >     - Additionally, NCAT trained models generalize well to unseen threat models, achieving near-SOTA robustness even on Perceptual Projected Gradient Descent (PPGD), which comprises one of the strongest attacks known to date.
> >
> > $ $
> >
> > We thank the reviewer again for their valuable comments and suggestions. We kindly ask if the reviewer would consider increasing their score to support acceptance of the paper if their concerns or questions have been addressed. We would also be glad to engage further during the author-reviewer discussion period.
> >
> > $ $
> >
> > [1] E. Wong, L. Rice and J. Z. Kolter, Fast is better than free: Revisiting adversarial training. In International Conference on Learning Representations (ICLR), 2020
> >
> > [2] G. Leclerc, A. Ilyas, L. Engstrom, S. Park, H. Salman and A. Madry, https://github.com/libffcv/ffcv
> >
> > [3] G. Sriramanan, S. Addepalli, A. Baburaj, and V. B. Radhakrishnan. Towards efficient and effective adversarial training. In Advances in Neural Information Processing Systems (NeurIPS), 2021

---

> ### Author Response · Authors · 2022-08-08
> **Author-Reviewer Discussion Period**
>
> Dear Reviewer TwcU, since the deadline for the author-reviewer discussion period is approaching soon, we kindly ask if you have any further questions. We would be glad to engage further during this discussion period.
>
> We kindly ask if the reviewer would consider increasing their score to support acceptance of the paper if their concerns or questions have been addressed adequately.

---

### Author Response · Authors · 2022-08-02
**A Note to All Reviewers**

We sincerely thank the reviewers for their time and valuable feedback on our work. We are glad that the reviewers appreciate the effectiveness of the proposed method in establishing the first successful demonstration of efficient single-step training against $\ell_1$ adversaries, its efficacious extension to the union of $\ell_p$ norm adversaries, scalability to large datasets and thorough empirical evaluations. We greatly appreciate the valuable comments and suggestions, and we will certainly try to incorporate these in the final version of the paper.

---

### Meta-Review · Area_Chair_T7Nq · 2022-08-23

**Recommendation:** Accept
**Confidence:** Less certain

**Metareview:**

This paper received mixed recommendations that range from borderline reject to strong accept. After examining all reviewers' comments, author rebuttal, and the paper itself, I lean toward acceptance mainly because: (i) it seems that the author responses have addressed reviewers' main concerns (although some reviewers did not respond to the rebuttal); (2) the contributions made by the paper, as summarized in the paper and author rebuttal, are significant enough and could potentially facilitate the development of adversarial defense.

Nevertheless, the organization and presentation of the paper needs to be further improved. For instance, the novelty/contributions over some existing works (e.g., [1] [2]), the computational cost of the proposed method, and the possible limitations are insufficiently discussed in the main text of the original submission. I urge the authors to consider these issues and take careful note of the reviewers' comments and suggestions when preparing for the final version.

[1] F. Croce and M. Hein. Adversarial robustness against multiple lp-threat models at the price of one and how to quickly fine-tune robust models to another threat model.
[2] G. Sriramanan, S. Addepalli, A. Baburaj, and V. B. Radhakrishnan. Towards efficient and effective adversarial training. In NeurIPS 2021


**Award:**

No

---

### Decision · Program_Chairs · 2022-09-14

Accept